# Facile Synthesis of Graphene from Waste Tire/Silica Hybrid Additives and Optimization Study for the Fabrication of Thermally Enhanced Cement Grouts

**DOI:** 10.3390/molecules25040886

**Published:** 2020-02-17

**Authors:** Ilayda Berktas, Ali Nejad Ghafar, Patrick Fontana, Ayten Caputcu, Yusuf Menceloglu, Burcu Saner Okan

**Affiliations:** 1Sabanci University Integrated Manufacturing Technologies Research and Application Center & Composite Technologies Center of Excellence, Teknopark Istanbul, 34906 Pendik, Istanbul, Turkey; ilayda.berktas@sabanciuniv.edu (I.B.); yusuf.menceloglu@sabanciuniv.edu (Y.M.); 2RISE Research Institutes of Sweden, Division Samhällsbyggnad–RISE CBI Betonginstitutet, Drottning Kristinas väg 26, 114 28 Stockholm, Sweden; ali.nejad.ghafar@ri.se (A.N.G.); patrick.fontana@ri.se (P.F.); 3Cimsa Cimento Sanayi A. S., Toroslar Mah. Tekke Cad., 33013 Yenitaskent, Mersin, Turkey; a.caputcu@cimsa.com.tr; 4Faculty of Engineering and Natural Sciences, Sabanci University, Orhanli, 34956 Tuzla, Istanbul, Turkey

**Keywords:** graphene nanoplatelet, waste tire, silanization, hybridization, thermal conductivity, grouts

## Abstract

This work evaluates the effects of newly designed graphene/silica hybrid additives on the properties of cementitious grout. In the hybrid structure, graphene nanoplatelet (GNP) obtained from waste tire was used to improve the thermal conductivity and reduce the cost and environmental impacts by using recyclable sources. Additionally, functionalized silica nanoparticles were utilized to enhance the dispersion and solubility of carbon material and thus the hydrolyzable groups of silane coupling agent were attached to the silica surface. Then, the hybridization of GNP and functionalized silica was conducted to make proper bridges and develop hybrid structures by tailoring carbon/silica ratios. Afterwards, special grout formulations were studied by incorporating these hybrid additives at different loadings. As the amount of hybrid additive incorporated into grout suspension increased from 3 to 5 *wt*%, water uptake increased from 660 to 725 g resulting in the reduction of thermal conductivity by 20.6%. On the other hand, as the concentration of GNP in hybrid structure increased, water demand was reduced, and thus the enhancement in thermal conductivity was improved by approximately 29% at the same loading ratios of hybrids in the prepared grout mixes. Therefore, these developed hybrid additives showed noticeable potential as a thermal enhancement material in cement-based grouts.

## 1. Introduction

In geothermal energy systems, the thermal conductivity of the grout used for backfilling the heat exchange boreholes and the pipes used in the loops for circulating the heat carrier fluid has been considered as an important issue for the improvement of the efficiency of the system. That is because the media influencing the heat exchange between the heat carrier fluid (in the loops) and the surrounding formations (i.e., soil or rock) include the pipe’s wall and the backfill materials in the borehole [1,2]. The poor thermal conductivity of neat cementitious grout not only decreases the efficiency of the system performance but also influences thermal cracking of the used backfill grout due to the high-temperature gradient between the pipe and the surrounding ground during the heat injection or extraction process [3]. Therefore, it is crucial to provide grouting materials with sufficiently improved thermal conductivity, while ensuring the other important properties such as the rheological properties, permeability, bleeding, and workability are in the accepted ranges. Accordingly, graphene is a promising candidate to incorporate into the grouting materials due to its high thermal conductivity property.

The integration of graphene-based materials in cement paste can reduce the porosity and the rate of hydration resulting in the development of stronger and more durable products [4]. In one of the recent studies, 0.01 *wt*% of graphene oxide (GO) nanosheets were mixed with cementitious materials consisting of ordinary Portland cement, silica fume, and ground granulated blast-furnace slag and increased the compressive strength of cement as about 7.82% after 28 days of curing [5]. Furthermore, Shang et al. demonstrated that using GO encapsulated silica fume, one can provide better rheological properties and increase the compressive strength of cement paste by 15.1% only by the addition of 0.04 *wt*% of GO [6]. Accordingly, previous studies are mostly focused on the enhancement of mechanical and rheological properties of cement by low loading graphene. However, there is limited work done on the introduction of graphene in cement-based materials to improve the thermal conductivity. For instance, Sedaghat et al. demonstrated that addition of 1% graphene did not have any significant effect on thermal diffusivity of the mixture, but incorporation of 5% graphene enhanced the thermal diffusivity by 25% at 25 °C and about 30% at 400 °C compared to that using the neat cement paste [4]. In another work, Ramakrishnan et al. incorporated 0.5 *wt*% of graphite, carbon nanotubes and graphene nanoplatelets (GNP) into form-stable phase change material-based composites and observed that using those additives led to the enhancement of thermal conductivity by 45%, 30%, and 49%, respectively [7].

One of the main factors that affect the thermal conductivity in cement paste/mortar is water/cement ratio, since increasing the water content reduces the density, increases the porosity that finally decreases the thermal conductivity [8,9]. Jobmann and Buntebarth showed that the water uptake decreased from 8.4% to 0.1% between 5% and 95% graphite and increased the thermal conductivity up to 3.67 W m^−1^K^−1^ with the composition of 10% graphite and 90% bentonite at 20 °C [10]. Herein, it is significant to adjust the water content between water-bearing bentonite and water-free graphite to attain high thermal conductivity, and thus surface chemistry of selected additives becomes a crucial factor in the mixing of cementitious materials.

Silanization has taken on special attention in the surface functionalization and the adjustment of hydrophilicity to control the penetration of water in cement structure [11,12]. Silane coupling agents have a significant influence on the dispersion of matrix and also affect the thermal, mechanical and physical properties of nanocomposites. There are numerous attempts for the modification of silica by organosilanes to connect to organic groups and act as a bridging component [13]. Especially silanol groups in silica have the ability to react with silane coupling agents and make the silica much more suitable for coupling reactions [14]. Among silane coupling agents, 3-aminopropyl triethoxysilane (APTES) was widely preferred as a binding agent in several applications such as composites, coatings, and adhesives [15]. Wang et al. reported that silane-modified GO polymerized with acrylic acid showed better distribution in saturated lime water than neat GO [16]. Zhao et al. stated that hybrid additive, which was produced by the impregnation of silica nanoparticles on GO modified by polycarboxylate superplasticizer, was added into cement matrix (1.5% SiO_2_ and 0.02% GO by weight of cement) and increased the compressive strength as about 38.31%, 44.47%, and 38.89% at 3th, 7th and 28th days, respectively [17].

Although numerous studies have been carried out to improve the different properties of cement pastes by using GO and modified GO sheets, there is still growing interest in the subject with the aim to reduce carbon footprint and develop sustainable and durable cement paste/grout. Herein, carbon-based materials obtained from waste sources such as gamma-irradiated recycled plastic [18], carbon powder waste obtained from the cutting process of laminate carbon composite [19] and rice husk ash [20] can be good alternatives to GO produced by harsh acidic and toxic conditions [21,22,23] to be used as an additive in grout mixtures. Another important issue is to reduce manufacturing costs by using carbon-by products or waste carbon materials. Therefore, a new methodology should be developed to address the issue related to thermal conductivity, aggregation, cost, and environmental impact of grouting.

In the present study, the main objective is to develop hybrid silica-GNP additives to enhance the thermal conductivity of the grout and thus increase the efficiency of the heat transmission and prevent the aggregation of treated hybrid additives in grout mixture and also decrease the manufacturing costs by using waste sources. To the best of our knowledge, there is no work about the utilization of graphene nanoplatelets produced from recycled carbon black obtained from the pyrolysis of waste tire as an additive in the preparation of grout. In order to prevent agglomeration and reduce the water absorption, silica particles were functionalized by APTES to make a suitable bridge with the surface of GNP. Then, the developed hybrid additives were added into the grout mixture by changing additive and water ratios, and the flow behaviors and the thermal conductivity property of the prepared grout mixtures were investigated in detail to monitor the effect of carbon content on the performance of the grouts.

## 2. Results and Discussion

### 2.1. Optimization Study for Surface Functionalization of Silica

Among silane coupling agents, 3-Aminopropyl triethoxysilane (APTES) is a widely used coupling agent. The chemical structure of APTES includes an amine functional group (-NH_2_) and three hydrolyzable groups which can be attached to the surface of silica. Figure 1 represents the reaction mechanism of silica functionalization by APTES schematically. Hydrolyzable groups of (-OCH_2_CH)_3_ in the structure of APTES was converted into -OH groups during hydrolysis, as shown in Figure 1a. After the condensation reaction occurred, pH was adjusted as 5.5, and then APTES was attached to the silica surface with different bridging modes, as seen in Figure 1b. In this step, NH_2_ groups of APTES remained available in the tails for the hybridization with GNP, whereas hydrolyzable groups were linked to the -OH groups on the surface of silica particles. In other words, APTES acts as a bridge between silica surface and carbon. An optimization study was conducted using three different APTES ratios to get an ideal surface composition.

First, amorphous silica having the surface area of 473.8 m^2^·g^−1^ with the particle size of 258 nm was selected, and three different Si:APTES ratios were studied for APTES functionalization. FTIR characterization was then carried out to identify the functional groups and observe the effect of APTES amount on the surface of silica. Figure 2 shows the FTIR spectra of APTES, neat silica, and APTES functionalized silica particles. Herein, the most prominent peaks for all spectra are located between 950 and 1250 cm^−1^ attributed to Si-O-Si and Si-O-C modes [24], and -OH bending vibration appeared at 800 cm^−1^ [25]. Furthermore, CH_2_ asymmetric and symmetric stretching modes that can be seen at around 2932 and 2864 cm^−1^, respectively, indicate the presence of the propyl chains of APTES [26]. The two labeled peaks appeared at around 1500 and 1600 cm^−1^ belonging to the NH_2_ scissor vibrations indicating the presence of the NH_2_ terminal group of APTES [27]. These peaks become more prominent as APTES concentration increases. In addition, in the FTIR spectrum of APTES, the double peaks at 2803 and 2970 cm^−1^ are attributed to stretching modes of CH_2_ [28]. However, -NH peak did not appear in the spectra of APTES-functionalized silica particles since the peak belonging to Si is dominant and the intensity of amine groups coming from APTES functionalization is significantly low. On the other hand, the attachment of NH_3_ on the surface of silica was confirmed by XPS characterization. Appendix A presents the XPS results of silica functionalized with APTES at different ratios. The results indicated that nitrogen content is comparably lower than the other elements of carbon, oxygen, and silicon on silica surface, and the highest nitrogen amount is attained by the ratio of Silica:APTES = 1:2.

In order to monitor the degradation behavior of functionalized silica samples, thermogravimetric analysis (TGA) analysis was conducted under nitrogen atmosphere. Figure 3 represents TGA curves of three different APTES functionalized silica samples. Neat silica showed the most stable behavior with a weight loss of 1.73%. In functionalized silica samples, the first weight loss between 50–120 °C comes from physically absorbed water molecules, and the second weight loss is attributed to the removal of chemically absorbed water between 120–200 °C [29]. Then, there is a significant weight loss between 200–600 °C due to the removal of organo-functional groups [30]. In addition, there are variations in the weight loss values of functionalized silica particles owing to the differences in functionalization degree. The characterization results confirmed the binding of silane groups on the silica surface.

### 2.2. Morphological and Structural Properties of Silica-GNP Hybrid Additive

The morphological properties of the produced GNP based hybrid additives were examined by using macroscopic techniques. Figure 4 represents SEM images of GNP, neat silica and Si:GNP = 1:5 and Si:GNP = 1:10 hybrid materials. As shown in Figure 4a, GNP has a layered structure. Figure 4c indicates that, after the introduction of silica nanoparticles on the surface of GNP, particles were distributed randomly and coated on the graphene plates. As Si ratio decreased, aggregation was diminished, and more homogenous structure was observed, as seen in Figure 4d. TEM image also supports the platelet structure with an average size of 50 nm observed in Figure 5a. Silica particles with an average size of 3 nm were observed in Figure 5b,c showing the homogenously distributed APTES functionalized silica particles on GNP.

XPS analysis was performed to investigate the surface chemical composition of the produced samples. Table 1 represents XPS characterization results of silica, Silica:APTES = 1:2, GNP, and hybrid additives of Si:GNP = 1:5 and Si:GNP = 1:10. GNP has a specific surface area of 131 m^2^.g^−1^ with a chemical composition of 87 at% carbon, 9.1 at% oxygen, 2 at% silicon, 0.5 at% iron, and others (S and Zn). With the incorporation of APTES-functionalized silica on GNP, carbon content increased and thus the concentrations of oxygen and nitrogen decreased in hybrid materials when compared to that in the Si:APTES = 1:2 sample. In comparison to neat GNP, nitrogen-based groups appeared in the developed additives after the hybridization with functionalized silica and silica amounts also increased. The XPS peaks of C1s, O1s, Si2p, and N1s for neat and hybrid samples are shown in the XPS survey scan spectra, as seen in Figure 6a. After the functionalization of silica particles by APTES, N1s peak was appeared in the spectrum of Si:APTES = 1:2 depicting the successful functionalization. After the hybridization of Si:APTES = 1:2 with GNP, N1s peak disappeared indicating the linkage of the amino group with graphene during the reaction. Figure 6b indicates the changes in C1s peak intensities of neat and hybrid samples. The existence of C1s binding energy values of 284.28, 284.08, and 284.38 eV for GNP, Si:GNP = 1:5, Si:GNP = 1:10 denotes the presence of sp^2^ hybridized C=C/C–C bonds [31]. Figure 6c indicates the changes in O1s peak intensities by showing the formation of the Si–O and C=O bond for the case of Si:GNP = 1:5 and Si:GNP = 1:10 is observed at the binding energy of 532 eV [32,33]. N1s spectra of Silica:APTES = 1:2, Si:GNP = 1:5, and Si:GNP = 1:10 was shown in Figure 6d. The broad N1s peak of Silica:APTES = 1:2 sample at 399 eV in belongs to NH_3_ group whereas the N1s binding energy of Si:GNP = 1:5 and Si:GNP = 1:10 shows a hydrogen-bonded NH_2_ group at 401.2 eV [33]. Furthermore, the C/O ratios of GNP and its hybrids of Si:GNP = 1:5 and Si:GNP = 1:10 were calculated as 2.5, 0.6, and 0.8, respectively, as shown in Appendix A. These results demonstrated the adjustment of hybrid additive composition by systematic optimization of silica and graphene contents.

Figure 7 shows the TGA curves of silica, GNP, and hybrid additives of Si:GNP = 1:5 and Si:GNP = 1:10. In this figure, weight loss of GNP as a function of temperature under nitrogen atmosphere was about 9 at% at 1000 °C due to the removal of surface oxygen groups. Both hybrid materials lost weight slightly owing to the elimination of surface functional groups. This weight loss was 6% at 1000 °C for Si:GNP = 1:5 and 8% for Si:GNP = 1:10. Finally, the results show that, as silica concentration was increased, the hybrid materials became more stable due to the highest thermal stability of silica.

Figure 8a represents Raman spectra of silica, GNP and hybrid additives of Si:GNP = 1:5 and Si:GNP = 1:10. GNP has two main Raman peaks of D and G appeared at 1342 and 1585 cm^−1^, respectively. The first peak named as D peak is related to the disorder degree of graphene samples, while the second one named as G peak attributes to the vibrational mode of sp^2^ carbon in graphitic materials [34]. There was no detected Raman peak in the analysis conducted on the neat silica. The defect density and crystallinity were then estimated using the intensity ratio of D peak to G peak (I_D_/I_G_) [35,36]. After hybridization, the I_D_/I_G_ ratios of the hybrid materials were changed. The disorder of Si:GNP = 1:5 was slightly increased, whereas the increase in GNP content in Si:GNP = 1:10 led to a decrease in I_D_/I_G_ ratio indicating a more ordered structure. Appendix A summarizes the Raman peak intensities ratios (I_D_/I_G_) and the crystallinity index of GNP and the hybrid additives of Si:GNP = 1:5 and Si:GNP = 1:10. Figure 8b shows the XRD patterns of silica, GNP and the hybrid additives of Si:GNP = 1:5 and Si:GNP = 1:10. XRD analysis was conducted to monitor the changes in crystallinity. GNP has a broad and less intense (002) peak at around at 2θ = 25°. Furthermore, the peak at 2θ = 35.8° belongs to the (311) reflection of Fe catalyst coming from the production process of graphene from waste tire. By the addition of silica particles, the peak at around 2θ = 25° becomes wider. However, there is no significant difference between the two hybrid additives, since GNP has a more prominent structure that suppresses the silica peak.

### 2.3. Grout Formulations by GNP Based Hybrid Additives and Their Characteristics 

The rheological properties of the grout used to backfill the heat-exchange boreholes are essential for several reasons. A grout with good rheological properties can provide good pumpability, less entrapped air, and consequently, lower permeability, less sensitivity to freeze, and thaw cycles and accordingly more durability and good thermal contact between the grout, the pipes and the surrounding underground formations that lead to higher thermal conductivity between the heat carrier fluid and the ground [37]. One of the most important parameters that affects the efficiency of the geothermal energy system is the thermal resistance of the heat-exchange boreholes that, in turn, depend on the thermal properties of the backfill [2]. The thermal resistivity of the grout, *R_g_* can be determined using the following Equation (1), where *S_b_* is the borehole shape factor and *λ_g_* is the thermal conductivity of the backfill grout in terms of [W·m^−1^K^−1^] [38]:(1)$$R_{g}=\frac{1}{S_{b}\cdot \lambda_{g}}$$

Since *R_g_* and *λ_g_* are reciprocals of one another, the minimum thermal resistance of the borehole means the maximum thermal conductivity in correlation with the shape factor enhancing the heat transfer rate between the heat carrier fluid and the Earth [39].

Table 2 summarizes the thermal conductivity results of the grout samples prepared by the addition of silica-GNP hybrid additives at different loadings. In the first trials, the thermal conductivity of GNP based cement sample was measured at three curing conditions on 7th, 14th, and 28th days. The test results at different curing days indicated that there was a slight increase in thermal conductivity values as curing time was kept longer, as given in Appendix A. As seen in Table 2, as the carbon content is increased in both the hybrid structure and the grout mixture, the water uptake is increased compared to the reference grout. The increase in the water demand in Si:GNP = 1:5 samples (that occurred due to the higher loadings of 1–5 *wt*%) decreased the thermal conductivity values from 2.373 to 1.816 W·m^−1^K^−1^. As the content of GNP was doubled in the hybrid additive (from Si:GNP = 1:5 to Si:GNP = 1:10), water demand of the grout mix was decreased and thus the thermal conductivity of the grout sample was increased from 1.816 to 2.341 W·m^−1^K^−1^ at 5 *wt*% loading which corresponds to 29% improvement. In addition to thermal conductivity, Table 2 shows a summary of the results obtained from the flowability tests (using a Marshcone and a flow table), the bleeding tests (using glass cylinders) as well as the density measurements (using a Mud-balance). In this study, the target values for the Marshcone time and the flow-table test were in the range of 80–120 sec and 20–30 cm, respectively. Similarly, the maximum accepted value for the bleeding and the minimum accepted value for the density were 2% and 1.3 g·cm^−3^, respectively. As seen in Table 2, all the test results obtained from the grout samples having silica-GNP hybrid additives were in the accepted ranges.

## 3. Materials and Methods

### 3.1. Materials

In this investigation, 3-Aminopropyl triethoxysilane (APTES, >98%, 0.946 g·mL^−1^) and acetic acid were purchased from Sigma-Aldrich, St. Louis, MO, USA. Amorphous silica (SiO_2_) was purchased from Merck, Germany. Graphene nanoplatelet (GNP) was obtained from pyrolyzed waste tire provided by NANOGRAFEN Co., Gebze, Kocaeli, Turkey. Two types of silica sands from Kumsan (30–35 AFS and 60–70 AFS), superplasticizer from Sika (SRMC-310S) and bentonite from Canbensan were used in the preparation of grout mixtures.

### 3.2. Method of Surface Functionalization of Silica

In silica functionalization, 1 gr of silica was dispersed in 50 mL distilled water via an ultrasonic homogenizer from Hielscher Ultrasonics at room temperature to provide homogeneous dispersion. Then, 1 mL APTES was added into the mixture by adjusting weight to weight ratio of silica and silane amounts and the pH level of the solution was adjusted to 5.5 by dropping acetic acid. In this process, APTES amount was approximately equal to silica amount. The as-prepared mixture was refluxed at 80 °C for 24 h. At the end of the reaction, filtration was performed by washing with water and ethanol twice. The filtrate was dried in an oven at 70 °C for 24 h. In order to get optimum functionalization degree, silica and APTES ratios were adjusted. Table 3 summarizes silica functionalization conditions with three different APTES ratios.

### 3.3. Hybridization of Functionalized Silica with GNP 

Surface-functionalized silica was used for the modification of the surface of GNP to attain better dispersion in grout mixture. In the hybridization step, 740 gr GNP was dispersed in 7400 mL distilled water to prepare colloidal suspension under the sonication process. Then, an aqueous solution with Silica:APTES in the amount of 74:148 weight % was added slowly into the GNP suspension. The reaction was performed through refluxing at 80 °C for 24 h. The resultant material was directly applied to the filtration process, and the material was easily separated from the water. Then, the material was kept in a vacuum oven at 70 °C for 24 h. For grouting formulations, two different GNP based hybrid additives encoded as H-GNP-1 and H-GNP-2 were developed by changing silica and GNP ratios of 1:5 and 1:10, respectively.

### 3.4. Preparation of Grouts by the Addition of Si-GNP Hybrid Additives 

In the preparation of grouts, water, superplasticizer (SP), and Si-GNP hybrid additives were mixed for 2 min at 2000 rpm using a high share mixer (VMA- Getzmann). Then, bentonite, cement and two types of silica sands were added orderly into the mixture and mixed at 6000 rpm for 4 min. Several experiments, including Marshcone test and flow-table test, were carried out to evaluate the developed grout flow properties. The developed grout was then molded in cylindrical molds (20 mm height and 60 mm diameter) and cured at 100% relative humidity and 20 °C for evaluation or their thermal conductivity.

### 3.5. Characterization

The morphological studies of GNP and its hybrid additives were analyzed using a Leo Supra 35VP field emission scanning electron microscope (SEM) and a JEOL JEM-ARM200CFEG UHR- transmission electron microscopy (TEM). X-ray diffraction (XRD) measurements were carried out by using a Bruker D2 PHASER Desktop with a CuKα radiation source. Raman spectroscopy was employed to characterize the structural changes in GNP samples using a Renishaw inVia Reflex Raman Microscopy System with a laser wavelength of 532 nm in the range of 100–3500 cm^−1^. Functional groups of functionalized silica samples were analyzed using a Thermo Scientific Fourier transform infrared spectroscopy (FTIR). The surface composition of GNP and its hybrid additives were examined quantitively by Thermo Scientific K-Alpha X-ray photoelectron spectrometer system (XPS). Zetasizer Nano ZS, Malvern dynamic light scattering (DLS) was used to measure the particle size of carbon and silica samples. Surface areas of the prepared samples were measured by BET method by using Micromeritics 3Flex equipment. Thermal conductivity analysis of GNP based grouts was conducted by hot disk thermal constants analyzer, TPS 2500 S. Thermogravimetric analysis (TGA) was carried out using a Mettler Toledo thermal analyzer (TGA/DSC 3+) over the temperature range of 25 °C to 1000 °C at a heating rate of 10 °K min^−1^ under nitrogen.

## 4. Conclusions

In the present study, silane functionalization routes were developed to treat silica surface and make compatible hybridization with GNP. Optimization study provided the proper amount of APTES (i.e., 1:2 of silica to APTES (*w*/*w*)) to be used in the treatment of silica. This was verified by FTIR and TGA analyses. Then, GNP produced from the recycled carbon black obtained by the pyrolysis of waste tire was selected as a carbon source for the hybridization step. This type of graphene has also surface oxygen functional groups of 9 at% to make suitable bridges with amine groups on the surface of APTES functionalized silica. After the structural confirmation of hybrid additive, reference grout formulation was determined by adjusting the contents of cement, silica sands, bentonite, superplasticizer, and water. The effects of GNP amount in hybrid structure and the concentration of hybrid additive on the thermal conductivity of the prepared grouts showed that as water content increased, thermal conductivity value decreased. On the other hand, increasing GNP amount in hybrid additive led to an increase in thermal conductivity by 29% by keeping the GNP loading ratio of 5*wt*% in two different grouts. Consequently, the study finally shows the noticeable potential of the hybrid additives produced from GNP to be used in the backfill grout formulations in the geothermal heat-exchange boreholes. This will be more discernible since renewable energy sources come into prominence by ever-increasing energy-demand and global pollution.

## Figures and Tables

**Figure 1 molecules-25-00886-f001:**
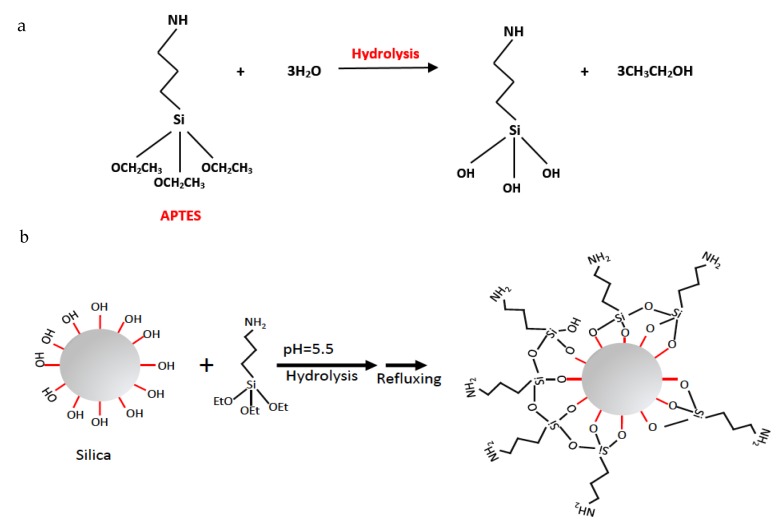
Schematic representation of the reaction of 3-aminopropyl triethoxysilane (APTES) functionalized silica particles in water: (**a**) Hydrolysis and (**b**) condensation reactions.

**Figure 2 molecules-25-00886-f002:**
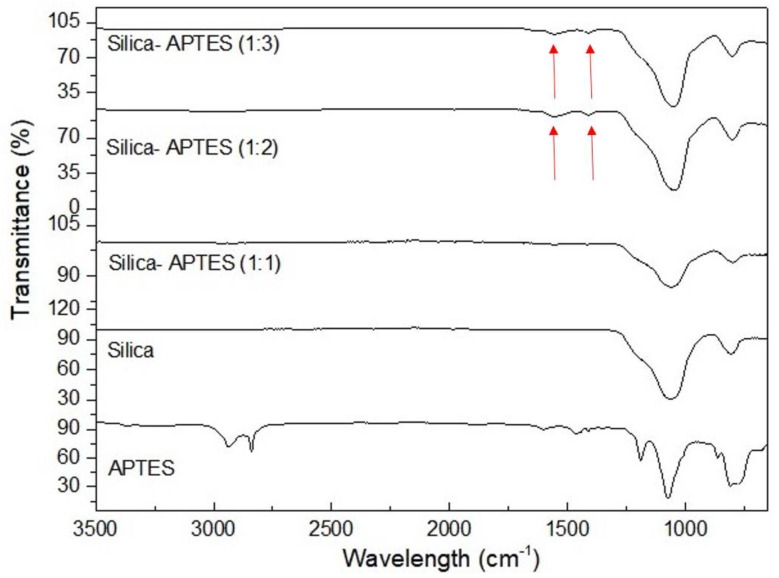
FTIR spectra of APTES, neat silica and APTES functionalized silica with three different ratios.

**Figure 3 molecules-25-00886-f003:**
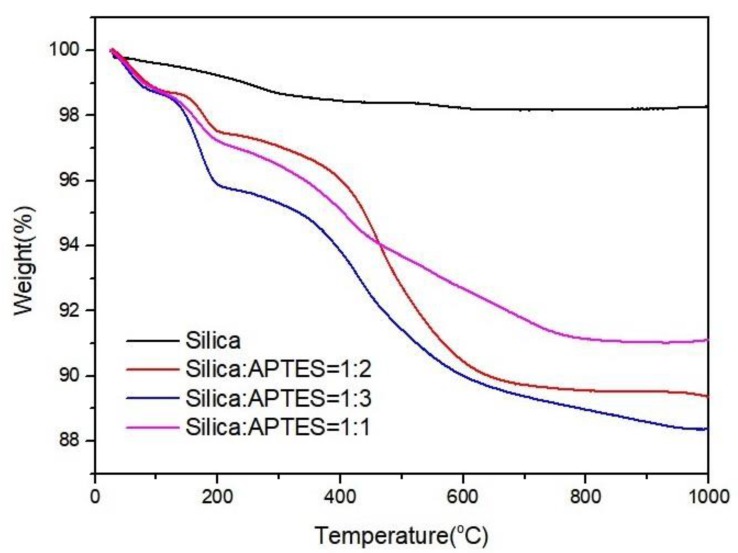
Thermogravimetric analysis (TGA) curves of APTES functionalized silica particles with different Si:APTES ratios.

**Figure 4 molecules-25-00886-f004:**
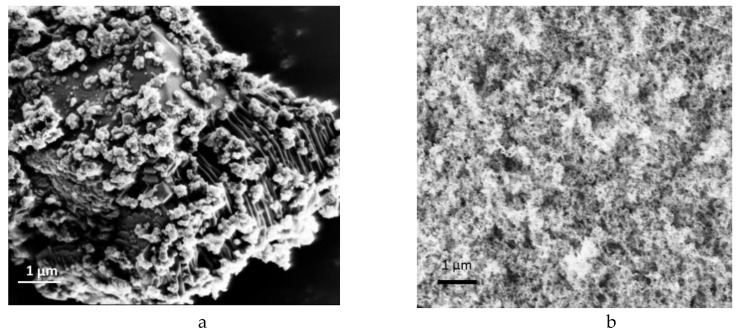
SEM images of (**a**) neat graphene nanoplatelet (GNP), (**b**) neat silica, (**c**) Si:GNP = 1:5 and (**d**) Si:GNP = 1:10 hybrid additives.

**Figure 5 molecules-25-00886-f005:**
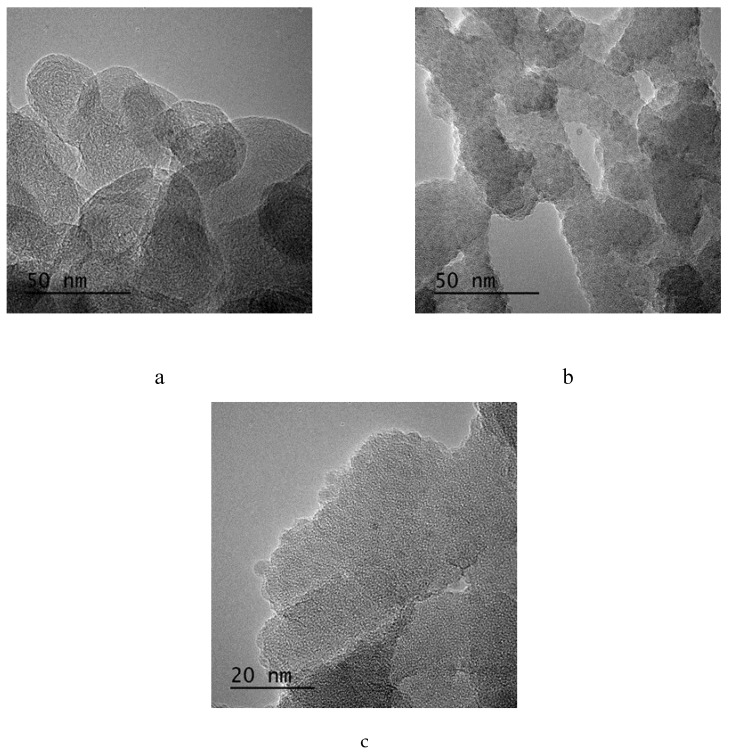
TEM images of (**a**) neat GNP, (**b**) and (**c**) its hybrid of Si:GNP = 1:10.

**Figure 6 molecules-25-00886-f006:**
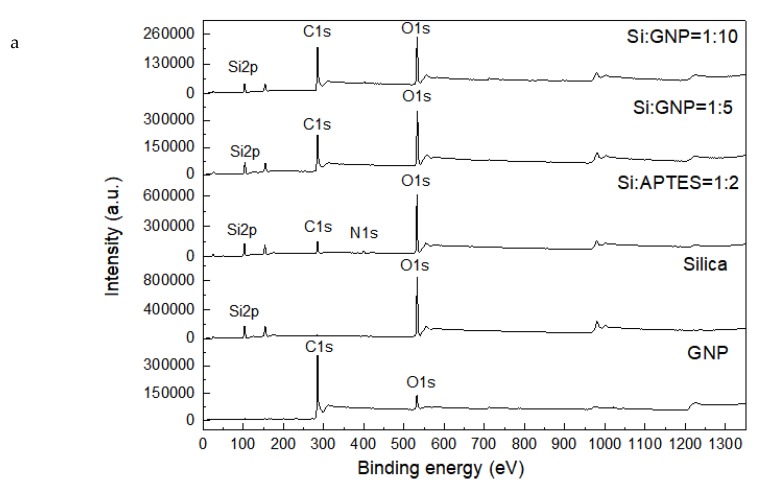
(**a**) XPS survey scan spectra, (**b**) C1s spectra, (**c**) O1s spectra and (**d**) N1s spectra of silica, GNP, Si:GNP = 1:5, Si:GNP = 1:10, and Silica:APTES = 1:2.

**Figure 7 molecules-25-00886-f007:**
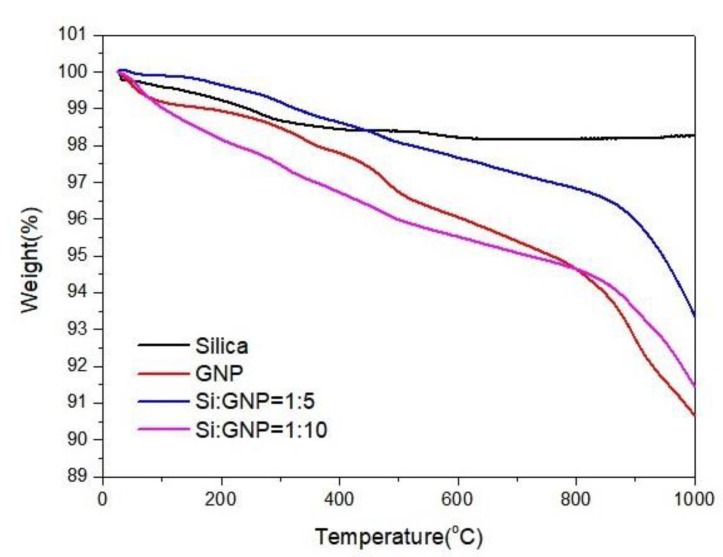
TGA curves of silica, GNP, Si:GNP = 1:5 and Si:GNP = 1:10 hybrid materials.

**Figure 8 molecules-25-00886-f008:**
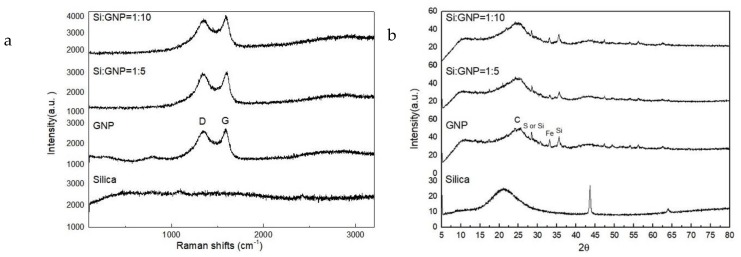
(**a**) Raman spectra and (**b**) XRD patterns of silica, GNP, and Si:GNP = 1:5 and Si:GNP = 1:10 hybrid additives.

**Table 1 molecules-25-00886-t001:** XPS results of GNP and its hybrids of Si:GNP = 1:5 and Si:GNP = 1:10.

Samples	Carbon (at%)	Oxygen (at%)	Silicon (at%)	Nitrogen (at%)	Other (at%)
**Silica:APTES = 1:2**	26	43	27	3.1	-
**GNP**	87	9	2	-	2
**Si:GNP = 1:5**	53	30	16	1	-
**Si:GNP = 1:10**	60	25.1	12.7	1.5	0.7

**Table 2 molecules-25-00886-t002:** Thermal conductivity results and rheological properties of selected reference grout and samples having hybrid additives.

Test Number	Cement (gr)	Silica Sand 30–35 AFS (gr)	Silica Sand 60–70 AFS (gr)	Bentonite (gr)	Additive (gr)	SP (gr)	Water (gr)	Marshcone (sec)	Flowtable (cm)	Bleeding (%)	Density (g·cm^−3^)	Thermal Conductivity (W·m^−1^K^−1^)
1	930	900	900	10	0 Reference	18.6	650	77	26	0.49	2.1	2.373
2	930	900	900	10	9.3 (Si:GNP = 1:5) (1 *wt*%)	18.6	650	90	28	<0.3	2.02	2.427
3	930	900	900	10	27.9 (Si:GNP = 1:5) (3 *wt*%)	18.6	660	105	24	0.10	2.04	2.287
4	930	900	900	10	46.5 (Si:GNP = 1:5) (5 *wt*%)	18.6	725	95	27	0.25	2.03	1.816
5	930	900	900	10	46.5 (Si:GNP = 1:10) (5 *wt*%)	18.6	700	96	28	1.2	2.05	2.341

**Table 3 molecules-25-00886-t003:** The reaction conditions of silica functionalization with three different APTES ratios.

Sample	Silica Amount (g)	APTES Amount (mL)	Reaction Time (h)	Reaction Medium	Reaction Temperature (°C)
Si:APTES = 1:1	1	1	24	Water	80
Si:APTES = 1:2	1	2	24	Water	80
Si:APTES = 1:3	1	3	24	Water	80

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
