# Peer review of "Facile Synthesis of Graphene from Waste Tire/Silica Hybrid Additives and Optimization Study for the Fabrication of Thermally Enhanced Cement Grouts"

_molecules, 2020, doi:10.3390/molecules25040886_

Round 1

Reviewer 1 Report

Manuscript is written in good level, and is focused on highly actual topic.  I have only few technical comments:

I recommend to unify  writting of units: page 4, line 125 ....473.8 m2.g-1... page 9, line 236 ....W.m-1.K-1.... page 9, line 256 ...g.cm-3....also  check writng  of units in tables and in whole document.  Table.3, Reaction Temperature (C°), correct is (°C) Figure 6a, Intensity % is correct for Y axis dessignation?

Author Response

The answers and explanations of 1st Reviewer’s comments:

We would like to thank the reviewer for his/her precious opinions and comments.

I recommend to unify writting of units: page 4, line 125 ....473.8 m .g ... page 9, line 236 ....W.m .K .... page 9, line 256 ...g.cm ....also check writng of units in tables and in whole document.

The whole document was reviewed and the units were revised to maintain integrity.

Table 3, Reaction Temperature (C°), correct is (°C)

The correction has been made in Table 3.

Sample

Silica amount

(g)

APTES amount (ml)

Reaction Time

(h)

Reaction

Medium

Reaction Temperature (°C)

Si:APTES=1:1

1

1

24

Water

80

Si:APTES=1:2

1

2

24

Water

80

Si:APTES=1:3

1

3

24

Water

80

Figure 6a, Intensity % is correct for Y axis designation?

The unit in Figure 6a has been corrected as Intensity (a.u.).

Reviewer 2 Report

The manuscript describes the characterization of silica nanomaterials and their interaction with GNPs. Here, IR spectroscopy, thermogravimetry, scanning electron microscopy, transmission electron microscopy, XPS, Raman spectroscopy and XRD are used. The focus is on the conductivity of the material when used in cement.

The concept of the manuscript is quite interesting. However, I think revisions will be needed concerning my following issues:

Abstract:

I miss an introductory part in the abstract. Why are you using GNPs? And why do you want to attach them to silica?

I enjoyed reading your introduction, where you explained these issues and I think you can extract part of it to your abstract.

Your °C looks strange thoughout the manuscript.

Discussion:

Figure 1: There is an error in the picture. The 3 of OCH2CH3 is missing. (There is only OCH2CH in the drawing of your APTES)

Figure 2: Why is there no OH or NH stretch vibration visible in all silica containing spectra?

Figure 6: Did you shift your XP spectra due to charging effects? Silica-Aptes looks a bit shifted lower binding energies for C1s and O1s spectra. This effect could occur due to charging effects since the conductivity of silica should not be so high.

Please add detail spectra oft he N1s region in order to observe the content of nitrogen in your samples.

How did you calculate the mass percent? Did you consider the transfer function of your instrument or did you just use the integrals of your peaks?

Figure 7: Why did you stop at 1000°C Higher temperatures for TG? Here it is getting interesting and you can observe a mass loss.

Line 199 due to the removal of surface oxygen groups

Why should you get rid of surface oxygen groups? How are they bound to GNP? Where do they originate from?

Table S2: Why do you calculate the intensities in counts? Usually, the integral should be considered. Furthermore, fort he D/G ratio, the D‘ band should be considered.

How do you calculate the crystallinity? I think the table misses a % as unit fort he crystallinity.

Table 2

How often did you repeat the conductivity tests? Since this is the focus of this investigation and the application of your material, you should test it multiple times.

In silica functionalization, 1 gr of silica was dispersed in 50 mL distilled water via Ultrasonic 273 Homogenizer from Hielscher Ultrasonics at room temperature to provide homogeneous dispersion.

How can you verify the dispersion?

The mixture was stirred through refluxing 277 overnight

At which temperature?

The preparation methods for all methods is missing

What heating rate did you choose for TGA and which device did you use?

Which method was used for IR spectroscopy? Transmission IR?

Author Response

The answers and explanations of 2nd Reviewer’s comments:
We would like to convey our sincere thanks to you and the referees for the positive and constructive comments, which we sincerely believe that these comments have improved both content and readability of manuscript significantly. The changes in the statements and new statements are highlighted in yellow in the revised manuscript.
ï‚® I miss an introductory part in the abstract. Why are you using GNPs? And why do you want to attach them to silica?
I enjoyed reading your introduction, where you explained these issues and I think you can extract part of it to your abstract.
We would like to thank the referee for his/her constructive comments. All changes were highlighted in yellow in manuscript. We reduced the number of words down to 200 in the abstract and revised by adding the following introductory sentences to explain the reasons of particle selection:
“This work evaluates the effects of newly designed graphene/silica hybrid additives on the properties of cementitious grout. In the hybrid structure, graphene nanoplatelet (GNP) obtained from waste tire was used to improve the thermal conductivity and reduce the cost and environmental impacts by using recyclable sources. Additionally, functionalized silica nanoparticles were utilized enhance the dispersion and solubility of carbon material and thus the hydrolyzable groups of silane coupling agent were attached to silica surface.”

ï‚® Your °C looks strange thoughout the manuscript.
The symbol of °C was corrected in the whole manuscript.
ï‚® Figure 1: There is an error in the picture. The 3 of OCH2CH3 is missing. (There is only OCH2CH in the drawing of your APTES)
The correction was made in the picture as follows:

ï‚® Figure 2: Why is there no OH or NH stretch vibration visible in all silica containing spectra?
-OH bending vibration appeared at around 800 cm−1 and the related explanation is added into the text by giving reference as seen below:
and OH bending vibration appeared at 800 cm−1 [23].
[23] Zhang, D.; Hegab, H.E.; Lvov, Y.; Snow, L.D.; Palmer, J. Immobilization of cellulase on a silica gel substrate modified using a 3 ‑ APTES self ‑ assembled monolayer. Springerplus 2016.
In FTIR spectrum of APTES, the double peaks at 2803 cm-1 and 2970 cm-1 are attributed to -NH2 stretching mode. However, -NH peak was not appeared in the spectra of APTES functionalized silica particles since the peak belonging to Si is dominant and the intensity of amine groups coming from APTES functionalization is significantly low.
On the other hand, the attachment of NH3 on the surface of silica surface was confirmed by XPS characterization. Table S1 added in supplementary document presents the XPS results of silica functionalized with APTES at different ratios. As seen in the table, nitrogen content is comparably lower than the other elements of carbon, oxygen and silicon on silica surface and the highest nitrogen amount is attained by the ratio of Silica:APTES=1:2.

Table S1. XPS results of neat silica and silica functionalized with APTES at different ratios
Sample name Carbon (at%) Oxygen (at%) Silicon
(at%) Nitrogen
(at%)
Silica 3.3 60 36.7 -
Silica:APTES=1:1 18 49 31 2
Silica:APTES=1:2 26 43 27 3.1
Silica:APTES=1:3 15 50 32 3

According to reviewer’s comments, Table S1 in supplementary document and the explanation below in the manuscript were added:
In addition, in the FTIR spectrum of APTES, the double peaks at 2803 cm-1 and 2970 cm-1 are attributed to -NH2 stretching mode. However, -NH peak was not appeared in the spectra of APTES functionalized silica particles since the peak belonging to Si is dominant and the intensity of amine groups coming from APTES functionalization is significantly low. On the other hand, the attachment of NH3 on the surface of silica surface was confirmed by XPS characterization. Table S1 in supplementary document presents the XPS results of silica functionalized with APTES at different ratios. The results indicated that nitrogen content is comparably lower than the other elements of carbon, oxygen and silicon on silica surface and the highest nitrogen amount is attained by the ratio of Silica:APTES=1:2.

ï‚® Figure 6: Did you shift your XP spectra due to charging effects? Silica-Aptes looks a bit shifted lower binding energies for C1s and O1s spectra. This effect could occur due to charging effects since the conductivity of silica should not be so high.
No, it is the raw data. The shifting in hybrid samples is a little bit lower than APTES functionalized silica since carbon content has been changed by the addition of silica. This is directly related to the changes in chemical composition.
ï‚® Please add detail spectra of the N1s region in order to observe the content of nitrogen in your samples.
N1s spectra of Si:GNP=1:10, Si:GNP=1:5 and Silica:APTES=1:2 were inserted in Figure 6 encoded as Figure 6d and the related explanation were added into the manuscript as given below:
N1s spectra of Silica:APTES=1:2, Si:GNP=1:5 and Si:GNP=1:10 was shown in Figure 9d. The broad N1s peak of Silica:APTES=1:2 sample at 399 eV in belongs to NH3 group whereas the N1s binding energy of Si:GNP=1:5 and Si:GNP=1:10 shows a hydrogen bonded NH2 group at 401.2 eV [30].
[30] Williams, E.H.; Schreifels, J.A.; Davydov, A.; Oleshko, V.P. Selective streptavidin bioconjugation on silicon and silicon carbide nanowires for biosensor applications Selective streptavidin bioconjugation on silicon and silicon carbide nanowires for biosensor applications. 2013.

Figure 6. (a) XPS survey scan spectra, (b) C1s spectra and (c) O1s spectra and (d) N1s spectra of Silica, GNP, Si:GNP=1:5, Si:GNP=1:10 and Silica:APTES=1:2
ï‚® How did you calculate the mass percent? Did you consider the transfer function of your instrument or did you just use the integrals of your peaks?
Here, we calculated atomic percentage of the data. The atomic % was calculated by the integrals of the peaks. An example for atomic % calculation can be seen in the table below:

ï‚® Figure 7: Why did you stop at 1000°C Higher temperatures for TG? Here it is getting interesting and you can observe a mass loss.
The equipment we used for TGA analysis was able to conduct up to 1000°C. Moreover, considering the application of final product, which is Si- GNP introduced grout, the material will not be exposed to high temperatures. Therefore, analysis was conducted between 25-1000 °C that is a typical measuring range for these kind of materials. In addition, we mainly focused on the observation of mass loss depending on the existence of functional groups, which occurs between 200°C and 600°C.
ï‚® Line 199 due to the removal of Surface oxygen groups. Why should you get rid of Surface Oxygen groups? How are they bound to GNP? Where do they originate from?
As received GNP has 9 at% of surface functional groups which comes from the treatment steps of recycled carbon obtained from the pyrolysis of waste tire. In TGA curve of GNP, oxygen groups were eliminated from the structure under inert atmosphere as temperature increased.
ï‚® Table S2: Why do you calculate the intensities in counts? Usually, the integral should be considered. Furthermore, for the D/G ratio, the D‘ band should be considered.
In the present study, we calculated the intensity in counts because the main aim was to investigate the disorderness of hybrid structures after hybridization process. On the other hand, it is possible to calculate the intensities of D and G peaks by using an integrated area ratio or peak height ratio [1-3]. The intensities of D and G peaks can be calculated with both using an integrated area ratio or peak height ratio [4]. In addition, the spectra of Si:GNP hybrid additives do not have D’ peak since the structure of GNP is similar to graphene oxide [5]. Therefore, we took ID/IG into consideration to monitor structural changes.

[1] Hong, J.; Park, M.K.; Lee, E.J.; Lee, D.; Hwang, D.S.; Ryu, S. Origin of new broad raman D and G Peaks in annealed graphene. Sci. Rep. 2013, 3, 1–5.
[2] Eckmann, A.; Felten, A.; Verzhbitskiy, I.; Davey, R.; Casiraghi, C. A multi-wavelenght Raman study on the nature of defects in graphene. Submitted.
[3] Ferrari, A.C.; Basko, D.M. Raman spectroscopy as a versatile tool for studying the properties of graphene. Nat. Nanotechnol. 2013, 8, 235–246.
[4] Ferrari, A.C. Raman spectroscopy of graphene and graphite: Disorder, electron-phonon coupling, doping and nonadiabatic effects. Solid State Commun. 2007, 143, 47–57.
[5] Marcano, D.C.; Kosynkin, D. V.; Berlin, J.M.; Sinitskii, A.; Sun, Z.; Slesarev, A.; Alemany, L.B.; Lu, W.; Tour, J.M. Improved Synthesis of Graphene Oxide. ACS Nano 2010, 4, 4806–4814.

ï‚® How do you calculate the crystallinity? I think the table misses a % as unit for the crystallinity.
The crystallinity percentage was calculated with the software of the XRD Bruker D2 PHASER Desktop, which is Bruker's Diffrac.Eva. It has an option to estimate the crystallinity of a sample based on the peak area to total diffractogram area.
According to reviewer’s comments, we added the unit of % in crystallinty in Table S2 in supplementary document.

Samples D peak
intensity
(a.u.) G peak
intensity
(a.u.) ID/IG Crystallinity
(%)
GNP 2670.6 2754.5 0.97 24.1
Si:GNP=1:5 3043.9 3100 0.98 27.8
Si:GNP=1:10 3894.2 4111.6 0.94 23.3

ï‚® Table 2: How often did you repeat the conductivity tests? Since this is the focus of this investigation and the application of your material, you should test it multiple times.
The critical days for cement samples are 7th, 14th and 28th days due to curing conditions. Initially, the conductivity measurements were carried out on the 7th, 14th and 28th days after the cement sample was poured into the mold and placed under 100% moist environment. The test results indicated that there is a slight increase in thermal conductivity as curing time was kept as longer as given in below Table. In addition, thermal conductivity of each sample was measured three times to obtain the most accurate result.
Sample name Thermal conductivity (W/(m∙K))
7th day 14th day 28th day
%5 GNP 0.821 0.755 0.709

According to reviewer’s comments, we added the related explanation in “2.3. Grout formulations by GNP based hybrid additives and their characteristics” in the manuscript and also the related table in supplementary document:
In the first trials, thermal conductivity of GNP based cement sample was measured at three curing conditions on 7th, 14th and 28th days. The test results at different curing days indicated that there was a slight increase in thermal conductivity values as curing time was kept as longer as given in Table S4.
Table S4. Thermal conductivity results of 5% GNP based cement at the curing days of 7th, 14th and 28th
Sample name Thermal conductivity (W/(m∙K))
7th day 14th day 28th day
5% GNP based cement 0.821 0.755 0.709

ï‚® In silica functionalization, 1 gr of silica was dispersed in 50 mL distilled water via Ultrasonic 273 Homogenizer from Hielscher Ultrasonics at room temperature to provide homogeneous dispersion. How can you verify the dispersion?
When the silica particles do not disperse homogeneously in the water, we detect the aggregation in the beaker. Mechanical mixing does not provide enough power to provide homogeneous dispersion and thus we preferred to work with probe sonicator and provide low silica concentration by increasing water content. Therefore, we got well-dispersed silica suspensions.
ï‚® The mixture was stirred through refluxing (277) overnight, At which temperature? The preparation methods for all methods is missing:

According to reviewer’s comments, we revised the mentioned sentence and added the following sentence in the manuscript as follows:
The as-prepared mixture was refluxed at 80°C for 24 h. At the end of reaction, filtration was performed by washing with water and ethanol twice. The filtrate was dried in oven at 70°C for 24 h.
ï‚® What heating rate did you choose for TGA and which device did you use?
Heating rate:10°C/min N2, Mettler Toledo TGA/DSC 3+. This information is also added to the manuscript in characterization:
Thermogravimetric Analysis (TGA) was carried out using a Mettler Toledo thermal analyzer (TGA/DSC 3+) over the temperature range of 25 °C to 1000 °C at a heating rate of 10°K min−1 under nitrogen.
ï‚® Which method was used for IR spectroscopy? Transmission IR?
Transmission IR was used in the analysis conducted with FTIR. The samples were also prepared as pellets with Potassium bromide (KBr) using Hydraulic Press but the peaks were appeared less than powder sample.

Round 2

Reviewer 2 Report

The manuscript has been improved.

However, there are some issues which need to be fixed.

I still do not understand, why no OH stretch vibration is visible.

Furthermore, the following vibrations are wrongly assigned:

" the double peaks at 2803 cm-1 and 2970 cm-1 are attributed to -NH2 stretching mode. "

the vibrations around 2803 and 2970cm-1 should be assigned to CH-stretch vibrations. NH-stretch vibrations should be around 3200-3500 cm-1

Author Response

We thank the reviewer for his/her constructive comments and have edited the manuscript to address their concerns.

I still do not understand, why no OH stretch vibration is visible. Furthermore, the following vibrations are wrongly assigned: " the double peaks at 2803 cm-1 and 2970 cm-1 are attributed to -NH2 stretching mode. " the vibrations around 2803 and 2970cm-1 should be assigned to CH-stretch vibrations. NH-stretch vibrations should be around 3200-3500 cm-1

-OH stretch vibration is also not visible in the spectrum of unmodified silica and thus we did not detect any OH peak after silanization.  In addition, the most prominent peaks for all spectra are located between 950 cm-1 and 1250 cm-1 attributed to Si-O-Si and Si-O-C modes and the other peaks are suppressed.

According to reviewer’s comments, the assignment in the double peaks at 2803 cm-1 and 2970 cm-1 was corrected in the manuscript and the edited sentence is also given below by the addition of new reference:  

the double peaks at 2803 cm-1 and 2970 cm-1 are attributed to stretching modes of CH2 [26].
